# Stress-Testing Byzantine Defenses under Data Heterogeneity

**Latifa Errami** [*]
College of Computing
Mohammed VI Polytechnic University
Benguerir, MA
`latifa.errami@um6p.ma`

**Hajar El Hammouti**
College of Computing
Mohammed VI Polytechnic University
Benguerir, MA
`hajar.elhammouti@um6p.ma`

**El Houcine Bergou**
College of Computing
Mohammed VI Polytechnic University
Benguerir, MA
`elhoucine.bergou@um6p.ma`

## Abstract

In this work, we focus on Byzantine-resilient distributed learning. While considerable efforts have been made to develop robust aggregations, the underlying threat model, especially under non-IID data distributions, remains under-explored. This imbalance may create a false sense of security about the effectiveness of current defenses. To address this gap, we revisit and calibrate existing Byzantine attacks to better reflect the challenges of leaning on heterogeneous data, enabling more realistic stress testing of defenses. Through systematic evaluation on standard benchmark datasets and using diverse partitioning strategies, we show that data heterogeneity provides adversaries with a larger leeway for model poisoning. We leverage this insight to critically evaluate existing defenses. Our findings underscore the need to assess robustness not only through defense design, but also through carefully calibrated and realistic threat models.[2]

## 1  Introduction

Distributed Machine Learning [40] has seen rapid adoption across a variety of real-world applications. As a result, there has been a growing interest in ensuring that ML algorithms are trustworthy. Threats against ML models span adversarial perturbations [34], backdoor attacks [32], data and model poisoning [10, 4] to arbitrary faults and reliability issues. In distributed learning these threats fall under Byzantine faults [31, 5]. This adversarial abstraction encompasses a general set of faulty behaviors exhibited by nodes in a distributed system, including both unintentional errors (e.g. software bugs, hardware failures) and active malicious behavior [3, 37, 14]. Although the feasibility of Byzantine attacks in real-world ML systems has been debated, examples show they can arise both deliberately [9] and inadvertently [39]. To that end, Byzantine-robust aggregation rules replace simple averaging at the server. These estimators protect model integrity even when some clients behave maliciously. Under independent and identically distributed (IID) data, defenses have been both theoretically and empirically proven to guarantee exact robustness [5, 38, 27, 11, 36, 6, 15]. In such settings, poisoned updates are easier to detect since updates from clients with similar data

---

[*]Corresponding Author

[2]An extended version of this work appears in *IEEE Access* [13].

39th Conference on Neural Information Processing Systems (NeurIPS 2025) Workshop: *Reliable ML from Unreliable Data.*

distributions are comparable [16], allowing even stealthy strategies [3, 37] to stand out. However, real-world collaborative learning typically involves non-IID data, wherein participants hold different data distributions. This complicates the task of detecting and defending against attacks, as updates naturally vary across participants. Although, numerous works have tackled this challenge, some rely on a validation data [7, 8, 28] at the server, which may not always be available. Others mitigate variance by mixing client updates either based on euclidean distance or randomly [2, 18]; however, this risks blending benign and malicious updates, potentially worsening performance. Finally, some works rely on penalties either as a regularization [23, 12] or as a way to downgrade malicious clients impact on model update. Conversely, DnC [29] uses SVD and dimensionality reduction to filter out suspected clients while alleviating the curse if dimensionality. Finally, the addition of momentum helps the defenses safeguard against time-coupled attacks [17]. In this work, we focus on the threat model under data heterogeneity. Although this is a realistic setting for collaborative learning, few attacks are explicitly designed for the non-IID case. For instance, [29] introduces the Min-Max and Min-Sum attacks, which solve an optimization problem to craft perturbations that allow Byzantine gradients to blend in with the honest majority while exceeding the distance of the farthest benign client. This effectively weaponizes heterogeneity to conceal poisoning. Intuitively, the farther the worst benign client is from the majority, the larger the perturbation an adversary can apply without detection. Although effective, these attacks require access to the honest clients' gradients at each iteration and involve solving an optimization problem, making them impractical in many real-world settings. The only other attack explicitly designed for the non-IID case is Mimic [18], where the attacker does not inject poisoned gradients but instead over-represents an honest client, biasing the model toward that client's distribution. While this leads to degraded global performance, it is not an active poisoning strategy. Together, these limitations highlight the need for better-designed attacks in the Byzantine heterogeneous setting.

In this work, we revisit and adapt existing attacks for non-IID data and show that current evaluations often underestimate the true strength of the adversary. While prior work such as [30, 19] critiques evaluation practices, it overlooks state-of-the-art provably robust defenses under data heterogeneity, such as Bucketing [18] and NNM [2]. Moreover, these critiques primarily focus on the exclusion of strong poisoning attacks [29, 3, 37]. Our study reveals a broader issue: evaluations not only exclude strong Byzantine attacks but also fail to calibrate attack strength appropriately for the challenges posed by data heterogeneity.

To address this, we revisit the threat model in Byzantine machine learning. Since we are interested in robustness under data heterogeneity and its impact on the performance of Byzantine defenses: (1) We experiment with Byzantine attacks originally designed for the IID case, tuning their hyperparameters, particularly those controlling perturbation strength, for the non-IID setting. We find that attacks such as ALIE [3] and IPM [37] can cause significantly more damage than previously reported when properly adapted to non-IID data. (2) Specifically, we demonstrate that data heterogeneity allows stronger perturbations to remain undetected, leading to degraded model performance. Moreover, the higher the degree of heterogeneity, the greater the perturbation an adversary can exploit without being flagged. Our findings highlight a critical but often overlooked insight: robustness depends not only on sophisticated defenses, but also on realistic threat models and evaluation protocols.

## 2  Evaluation

Our goal is to investigate how well robust defenses withstand poisoning attacks under increasing levels of data heterogeneity, rather than by simply increasing the number of adversaries. While prior work typically stresses defenses by raising the proportion of Byzantine clients, this may be unrealistic in practice. As highlighted in [30, 35], real-world federated learning deployments often involve a small fraction of compromised clients. Instead, we stress-test defenses by leveraging a system property that is both uncontrollable and often overlooked: data heterogeneity.

We argue that more attention should be given to the severity of heterogeneity that may naturally arise in realistic settings, and that this should be explicitly reflected in robustness evaluations. From the attacker's perspective, greater heterogeneity enables stronger perturbations to remain undetected. To investigate this, we vary the strength parameters $z$ (for ALIE) and $\epsilon$ (for IPM) under a fixed adversarial budget. Specifically, we evaluate each defense under ALIE and IPM attacks with a fixed number of Byzantine clients, $b = 5$ out of $n = 25$, while varying both the attack strength ($z, \epsilon \in 0, 0.5, 2.5, 5, 8, 10$) and the heterogeneity level ($\beta \in 0.1, 0.3, 0.5$). Notably, prior evaluations

often use default attack strengths calibrated for stealth in IID settings, typically $\epsilon = 0.1$ for IPM and $z$ ranging from 0.25 to 2.5 for ALIE. In contrast, our study explores how much more damaging these attacks can become when properly adapted to heterogeneous data.

**Top-1 Test Accuracy**    We evaluate all models using Top-1 test accuracy. Reported values are the mean of three independent runs with distinct random seeds. Test accuracy is the standard metric in Byzantine-robust learning because it directly measures a defense's ability to preserve predictive performance under training-time poisoning. Since experiments are performed on widely used benchmark datasets with well-established baselines, any substantial drop in accuracy unambiguously signals that the defense has failed.

**Datasets & Data Splits**    We use three standard benchmark datasets in distributed learning: FMNIST (with LeNet-5 [22]), SVHN [26], and CIFAR-10 [20] (with AlexNet [21]). To simulate non-IID data, we skew the label distribution across clients using Latent Dirichlet Sampling, which draws client-specific label distributions from a Dirichlet distribution $Dir(\beta)$. The concentration parameter $\beta$ controls the degree of heterogeneity: smaller values of $\beta$ produce more imbalanced, heterogeneous data splits [24, 33, 2].

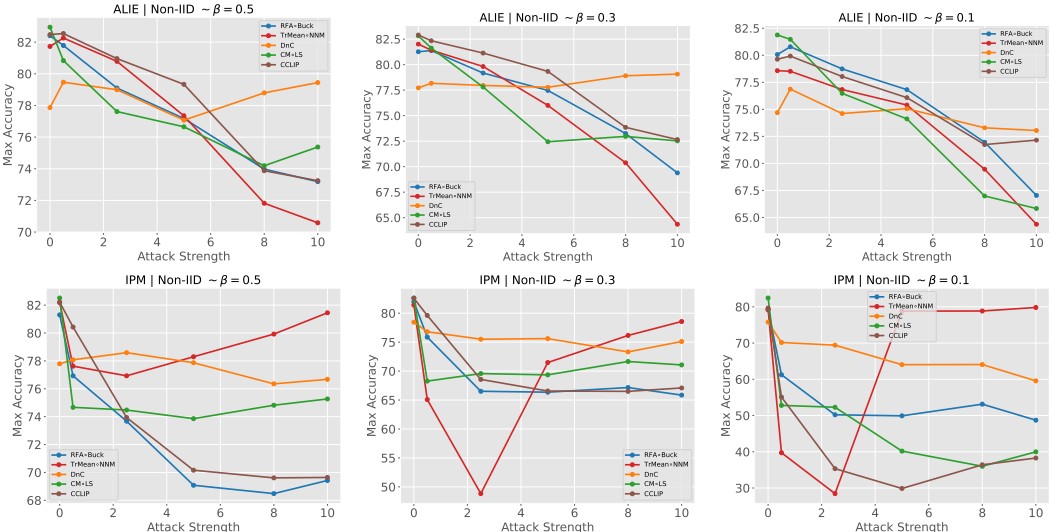

Figure 1: Maximum achieved $Top-1$ Test Accuracy for all studied aggregations on varying **FMNIST** non-IID splits. under $\delta = 20\%$ Byzantine performing for varying degrees of attack strength $z$ and $\epsilon$ for ALIE (Row 1) and IPM (Row 2) respectively

Table 1: Max (%) Top-1 Test Accuracy (mean±std) across $T = 6000$ for SVHN trained with 5 different defenses (averaged for 3 runs). We experiment with multiple levels of heterogeneity $\beta$. We have a proportion of malicious attackers $\delta = 17\%$. Each block, (i.e., for a dateset under a level of heterogeneity $\beta$), we **bold** the attacks for which the defense achieves the worst accuracy (i.e., the most potent attack)

|  |  | RFA(buck) | CMLS | CCLIP | trMean(NNM) | DnC |
|---|---|---|---|---|---|---|
| SVHN ($\beta = 0.5$) | ALIE ($z = 8$) | **26.34 ± 12.22** | **25.66 ± 8.58** | **29.97 ± 9.11** | **20.09 ± 0.71** | 87.55 ± 0.33 |
|  | IPM ($\epsilon = 2.5$) | 62.43 ± 28.73 | 84.32 ± 0.63 | 83.81 ± 1.45 | 83.41 ± 0.80 | 86.93 ± 0.36 |
|  | MinMax | 81.80 ± 1.61 | 81.56 ± 2.71 | 63.54 ± 16.57 | 81.82 ± 0.90 | 86.38 ± 1.21 |
|  | MinSum | 84.50 ± 2.15 | 83.55 ± 0.76 | 85.54 ± 0.54 | 84.23 ± 0.40 | 86.78 ± 0.71 |
|  | SF | 86.33 ± 0.32 | 78.48 ± 8.00 | 87.25 ± 0.35 | 82.78 ± 0.61 | **85.60 ± 0.97** |
| SVHN ($\beta = 0.3$) | ALIE ($z = 8$) | **26.15 ± 9.20** | **20.34 ± 1.06** | **27.30 ± 8.07** | 19.59 ± 0.00 | 87.82 ± 0.59 |
|  | IPM ($\epsilon = 2.5$) | 78.58 ± 1.26 | 80.29 ± 0.06 | 80.38 ± 2.19 | **16.29 ± 4.66** | 84.26 ± 0.91 |
|  | MinMax | 82.62 ± 0.52 | 79.19 ± 0.26 | 50.76 ± 29.41 | 79.59 ± 1.16 | **82.46 ± 1.47** |
|  | MinSum | 83.78 ± 0.46 | 79.60 ± 0.66 | 86.49 ± 0.47 | 82.26 ± 0.75 | 83.81 ± 1.60 |
|  | SF | 84.06 ± 1.20 | 80.15 ± 0.66 | 86.28 ± 0.09 | 39.43 ± 14.59 | 84.49 ± 0.30 |

Table 2: Top-1 Test Accuracy (%) (mean±std averaged for 3 runs) of different defenses trained for $T = 8000$ under various combinations of $\beta$ on CIFAR10 for $b = 3$ Byzantine clients out of $n = 17$.

|  |  | RFA(buck) | CMLS | CCLIP | trMean(NNM) |
|---|---|---|---|---|---|
| CIFAR10 ($\beta = 0.3$) | ALIE ($z = 8$) | **23.26 ± 1.39** | **21.71 ± 2.81** | **17.89 ± 5.64** | **21.93 ± 1.49** |
|  | IPM ($\epsilon = 2.5$) | 39.30 ± 0.35 | 44.20 ± 1.16 | 50.95 ± 1.53 | 52.29 ± 0.22 |
|  | MinMax | 37.99 ± 1.34 | 42.96 ± 0.68 | 37.81 ± 0.95 | 49.60 ± 0.87 |
|  | MinSum | 46.00 ± 0.50 | 43.36 ± 0.89 | 53.33 ± 0.41 | 52.67 ± 0.50 |
|  | SF | 52.76 ± 1.03 | 44.12 ± 0.43 | 63.58 ± 0.47 | 51.58 ± 2.03 |
| CIFAR10 ($\beta = 0.5$) | ALIE ($z = 8$) | **21.29 ± 2.15** | **19.01 ± 0.47** | **24.07 ± 1.43** | **18.90 ± 6.39** |
|  | IPM ($\epsilon = 2.5$) | 43.80 ± 1.39 | 50.53 ± 0.18 | 51.01 ± 0.90 | 56.55 ± 0.74 |
|  | MinMax | 44.26 ± 0.69 | 48.92 ± 2.36 | 35.33 ± 2.28 | 53.84 ± 0.07 |
|  | MinSum | 51.57 ± 1.14 | 49.05 ± 0.30 | 55.58 ± 0.89 | 59.78 ± 1.36 |
|  | SF | 53.37 ± 0.42 | 50.07 ± 2.24 | 64.62 ± 1.04 | 23.93 ± 9.05 |

**Failure Under Strong Perturbations.** The results presented in Figure 1 illustrate that current SoTA non-IID Byzantine defenses struggle under strong adversarial perturbations across heterogeneity levels. For the FMNIST dataset under ALIE attack (row 1 of figure 1) all defenses experience significant accuracy drops that grow as the perturbation increases, signaling that although the attack is getting aggressive the defenses are unable to filter out the poisoned updates. For MNIST however, DnC is capable of withstanding the poisoning maintaining accuracy comparable to the honest setting where no attacker is present and trMean(NNM) becomes effective recovering its original accuracy once the strength of the attack surpasses a threshold $z = 5$ especially under mild non-IID data ($\beta \in \{0.3, 0.5\}$).

Under IPM attack (Row 2 of figure 1) the same trend appears. Mainly, all defenses experience accuracy drops as the strength of the attack augments and as heterogeneity grows, however the decline in performance is not as aggressive as ALIE and most defenses accuracy plateaus once $\epsilon > 2.5$ at the exception of trMean(NNM) that always recovers its accuracy especially as the perturbation grows and heterogeneity drops. That is, as the attacker is becoming aggressive NNM succeeds at filtering out all poisoned updates. For the rest of the defenses, accuracy remains low as perturbations grow, that can be attributed to the way these defenses operate: RFA(Buck) randomly mixes gradients in the pre-processing phase, leading to contamination of honest gradients with highly poisoned updates making it harder to recover. The CMLS variant on the other hand may fail to recover because of its inclusion strategy. The LS defense include all submissions in the update with a penalty. Consequently, due to the large perturbations introduced by the attackers in this case the model becomes compromised as penalties may fail to contain the poisoned vectors.

**Defense Recommendation** Singular-value decomposition (SVD) coupled with dimensionality-reduction techniques, as used in **DnC** [29], has demonstrated strong empirical robustness across datasets and threat models. Yet Fig. 2 shows that DnC is also the slowest defense we evaluated, taking roughly 3 s per round at $d = 500k$ and $n = 25$, compared with 0.2 s for Buck (RFA with bucketing) and CMLS. Although sub-sampling $k \ll d$ gradient coordinates cuts the full-SVD cost of $\mathcal{O}(d^3)$, it however does not fully eliminate the overhead. Future work should explore lightweight, scalable spectral defenses that preserve robustness without prohibitive runtime. These results also highlight a broader limitation: similarity and distance-based aggregation strategies can fail under data heterogeneity, where honest updates naturally diverge.

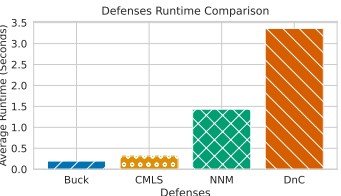

Figure 2: Runtime Comparison of the studied defenses with $n = 25$ and $d = 500k$.

**Comparing Attacks** We compare state of the art Byzantine attacks MinMax, MinSum that solve an optimization problem to find the optimal perturbation and SF [1], to ALIE with ($z = 8$) and IPM with ($\epsilon = 2.5$) the recovered optimal perturbations from our ablation studies. We test the studied defenses on SVHN table 1 and CIFAR10 table 2 under Dirichlet non-IID splits $\beta \in \{0.3, 0.5\}$. Across all settings, calibrated ALIE and IPM consistently degrade test accuracy more effectively than other

attacks. In particular, the highest Top-1 accuracy achieved under these calibrated attacks does not exceed $24\%$, highlighting their potency when tuned for heterogeneity.

## 3   Conclusion

In this work, we revisited the evaluation of Byzantine-robust defenses under data heterogeneity and revealed that existing practices systematically underestimate adversarial strength. By adapting classical IID-based attacks such as ALIE and IPM to the non-IID setting, we showed that state-of-the-art defenses can suffer substantial accuracy degradation once perturbations are properly calibrated to heterogeneity. These findings emphasize that robustness in distributed learning depends not only on sophisticated defenses, but also on realistic threat models and evaluation protocols that reflect the challenges of real-world deployments.

Looking ahead, advancing Byzantine-robust machine learning will require principled defenses that scale to large federated systems while withstanding calibrated, heterogeneous adversaries. Equally important will be the development of standardized benchmarks and threat models to foster reliable, reproducible evaluation practices in robust learning.

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

# A  Ethics considerations

This paper provides a re-evaluation of Byzantine robustness in distributed machine learning systems, uncovering how attacks can be optimized to exploit data heterogeneity, resulting in significant degradation of model accuracy. While detailing these vulnerabilities could potentially inform malicious actors, our primary objective is to highlight the limitations of current evaluation methodologies, which may offer a false sense of security.

By exposing these weaknesses, we aim to motivate the development of stronger defenses and ensure models are safeguarded against worst-case scenarios. In keeping with ethical research practices, it is crucial to balance the open dissemination of findings with the responsibility to prevent misuse.

While our work primarily focuses on identifying vulnerabilities, we underscore the importance of concurrently developing and implementing effective defense mechanisms to mitigate these risks.

By fostering transparent discussions about these issues, we contribute to the advancement of secure and resilient distributed learning systems, aligning with the broader goal of promoting ethical practices in machine learning research.

# B  Byzantine Attacks

In this section, we go over state of the art Byzantine attacks. We mainly consider those that are shown to incur significant damage to the accuracy achieved by distributed learning algorithms. We exclude simpler attacks namely Label Flipping (LF) [5], Gaussian Noise [25] and adaptive attacks that rely on knowledge about the aggregation [14] used by the server as they both acquire additional knowledge and are only powerful against a handful of defenses they are tailored for.

Byzantine attacks typically involve the manipulations of gradient vectors, rather than submitting random noise. This strategy allows adversaries to blend in with honest clients while still disrupting training dynamics. Let $\kappa \in \mathbb{R}^+$, $\eta \in \mathbb{R}$, $v \in \mathbb{R}^d$ and $\hat{g}$ a legitimate gradient vector either computed or intercepted by malicious clients. Most Byzantine attacks fall into one of the following categories:

- **Magnitude of the gradient**: This is a class of attacks that aims to poison the model by tampering with the magnitude of the gradient update, either by shifting or scaling the gradient. Attack vectors from this class can be written in the form $\kappa \hat{g} + \eta v$.

- **Direction of the Descent (Sign Inversion)**: Malicious clients invert the sign of their gradients (often sending $-\kappa \hat{g}$) to push the global update in the opposite direction. A simple yet disruptive attack causing the model to move away from the true descent direction.

- **Defense Manipulation**: These attacks take advantage of the learning setting and the way standard aggregations operate. In particular, the goal is to circumvent defenses [14, 18].

Table 3: Summary of prominent Byzantine attacks used in distributed learning. Each attack is characterized by its knowledge assumptions (e.g., omniscient vs. non-omniscient), whether it is aware of the aggregation rule, the form of its attack vector, whether collusion between adversaries is required, and its manipulation strategy (e.g., direction of the update, magnitude of the update, or targeted (tailored)). This classification highlights key differences in how attacks operate and the assumptions they make, which is essential for evaluating their practical applicability under various threat models.

| Attack | Knowledge | Attack Vector | Collusion | Manipulation Strategy |
|--------|-----------|---------------|-----------|------------------------|
| ALIE [3] | Non-omniscient Aggregation-agnostic | $g_b = \overline{g}_{\mathcal{B}} - z\,\sigma_{\mathcal{B}}$ | ✓ | Magnitude |
| IPM [37] | Non-omniscient Aggregation-agnostic | $g_b = -\frac{\epsilon}{|\mathcal{B}|}\sum_{b\in\mathcal{B}}\nabla\mathcal{L}_b(\theta_t)$ | ✓ | Direction |
| SF [1] | Non-omniscient Aggregation-agnostic | $g_b = -\nabla\mathcal{L}_b(\theta_t)$ | ✗ | Direction |
| Min-Max [29] | Non/Omniscient (Aggregation-aware) | $g_b = \overline{g}_{\text{ref}} + \gamma\nabla^p$ | ✓ | Magnitude |
| Min-Sum [29] | Non/Omniscient (Aggregation-aware) | $g_b = \overline{g}_{\text{ref}} + \gamma\nabla^p$ | ✓ | Magnitude |
| Mimic [18] | Omniscient Aggregator-agnostic | $g_b = \nabla\mathcal{L}_{i^*}(\theta_t)$ | ✓ | Defense-targeted |

