# OpenReview forum: "Stress-Testing Byzantine Defenses under Data Heterogeneity"
_NeurIPS.cc/2025/Workshop/Reliable_ML — NeurIPS 2025 - Reliable ML Workshop_

### Official Review · Reviewer_7zcr · 2025-09-19
**Byzantine defenses under non-IID data**

**Rating:** 10
**Confidence:** 4

**Review:**

Summary: This paper provides an analysis to understand how well the defenses against Byzantine attacks hold up in settings with heterogeneous data. Specifically, they look at attacks made for IID data, adapt them to non-IID data, and see how well they hold up. The implications of this work are that realistic evaluations are critical to stress test distributed learning systems.

Strengths: This work is very relevant for this workshop, and provides a brief yet powerful summary on issues that arise when dealing with heterogeneous data. The results are presented very clearly, and effectively motivates the overall message that better defenses and evaluation will be necessary when dealing in adversaries in heterogeneous settings. They also show that well-calibrated and adapted attacks can harm performance accuracy quite severely. Overall. a great, well-written paper, and will be relevant for much of the future research in Byzantine-resilient distributed learning.

---

### Official Review · Reviewer_8PoH · 2025-09-19
**Summary and review feedback**

**Rating:** 6
**Confidence:** 2

**Review:**

This paper investigates the robustness of Byzantine-resilient distributed learning under non-IID (heterogeneous) data distributions. While many existing defenses (e.g., RFA, Trimmed Mean, Krum variants, DnC) have been shown to work under IID assumptions, their robustness under heterogeneous data is less explored. The authors adapt classical Byzantine attacks — ALIE and IPM — originally designed for IID data, and recalibrate them for heterogeneous settings.
The authors show that heterogeneity amplifies adversarial strength: stronger perturbations remain undetected as natural variation between honest client updates masks poisoning.
The authors conduct systematic evaluation on FMNIST, SVHN, and CIFAR-10 using non-IID splits generated via Dirichlet distributions with varying concentration parameters.
Through the experiment, the authors demonstrate that current defenses fail under calibrated non-IID attacks, with significant drops in test accuracy. For instance, under strong ALIE/IPM attacks tuned for heterogeneity, accuracies collapse below 25\%.
However, this paper mainly adapts existing attacks (ALIE, IPM) rather than proposing fundamentally new ones. Therefore, I believe the contribution of this paper is more evaluation-based than algorithmic.
Additionally, the computational cost of different defenses method has not been quantified. While spectral defenses (e.g., DnC) are highlighted as robust but expensive, no concrete runtime or scalability benchmarks are provided.